# HCV Genetic Diversity Can Be Used to Infer Infection Recency and Time since Infection

**DOI:** 10.3390/v12111241

**Published:** 2020-10-31

**Authors:** Louisa A. Carlisle, Teja Turk, Karin J. Metzner, Herbert A. Mbunkah, Cyril Shah, Jürg Böni, Michael Huber, Dominique L. Braun, Jan Fehr, Luisa Salazar-Vizcaya, Andri Rauch, Sabine Yerly, Aude Nguyen, Matthias Cavassini, Marcel Stoeckle, Pietro Vernazza, Enos Bernasconi, Huldrych F. Günthard, Roger D. Kouyos

**Affiliations:** 1Division of Infectious Diseases and Hospital Epidemiology, University Hospital Zurich, CH-8091 Zurich, Switzerland; louisa.anja@hotmail.co.uk (L.A.C.); turk.teja@virology.uzh.ch (T.T.); Karin.Metzner@usz.ch (K.J.M.); afegenwimbunkah@gmail.com (H.A.M.); dominique.braun@usz.ch (D.L.B.); Jan.Fehr@usz.ch (J.F.); 2Institute of Medical Virology, University of Zurich, CH-8057 Zurich, Switzerland; shah.cyril@virology.uzh.ch (C.S.); boeni.juerg@virology.uzh.ch (J.B.); huber.michael@virology.uzh.ch (M.H.); 3Swiss National Reference Center for Retroviruses, University of Zurich, CH-8057 Zurich, Switzerland; 4Department of Public Health, Epidemiology Biostatistics and Prevention Institute, University of Zurich, CH-8001 Zurich, Switzerland; 5Department of Infectious Diseases, Bern University Hospital, University of Bern, CH-3010 Bern, Switzerland; luisapaola.salazarvizcaya@insel.ch (L.S.-V.); Andri.Rauch@insel.ch (A.R.); 6Laboratory of Virology, Division of Infectious Diseases, Geneva University Hospital, University of Geneva, CH-1205 Geneva, Switzerland; Sabine.Yerly@hcuge.ch (S.Y.); aude.nguyen@hcuge.ch (A.N.); 7Division of Infectious Diseases, Lausanne University Hospital, CH-1011 Lausanne, Switzerland; Matthias.Cavassini@chuv.ch; 8Division of Infectious Diseases and Hospital Epidemiology, University Hospital Basel, University of Basel, CH-4031 Basel, Switzerland; Marcel.Stoeckle@usb.ch; 9Division of Infectious Diseases, Cantonal Hospital St Gallen, CH-9007 St. Gallen, Switzerland; pietro.vernazza@kssg.ch; 10Division of Infectious Diseases, Regional Hospital Lugano, CH-6900 Lugano, Switzerland; Enos.bernasconi@eoc.ch

**Keywords:** hepatitis C virus infection, infection recency, genetic variation, sequence analysis, viral genomics

## Abstract

HIV-1 genetic diversity can be used to infer time since infection (TSI) and infection recency. We adapted this approach for HCV and identified genomic regions with informative diversity. We included 72 HCV/HIV-1 coinfected participants of the Swiss HIV Cohort Study, for whom reliable estimates of infection date and viral sequences were available. Average pairwise diversity (APD) was calculated over each codon position for the entire open reading frame of HCV. Utilizing cross validation, we evaluated the correlation of APD with TSI, and its ability to infer TSI via a linear model. We additionally studied the ability of diversity to classify infections as recent (infected for <1 year) or chronic, using receiver-operator-characteristic area under the curve (ROC-AUC) in 50 patients whose infection could be unambiguously classified as either recent or chronic. Measuring HCV diversity over third or all codon positions gave similar performances, and notable improvement over first or second codon positions. APD calculated over the entire genome enabled classification of infection recency (ROC-AUC = 0.76). Additionally, APD correlated with TSI (R^2^ = 0.33) and could predict TSI (mean absolute error = 1.67 years). Restricting the region over which APD was calculated to *E2*-*NS2* further improved accuracy (ROC-AUC = 0.85, R^2^ = 0.54, mean absolute error = 1.38 years)**.** Genetic diversity in HCV correlates with TSI and is a proxy for infection recency and TSI, even several years post-infection.

## 1. Introduction

Inferring the duration of infection is of key importance for understanding both the epidemiology and pathogenesis of hepatitis C virus (HCV) infections. From an epidemiological perspective, the time of infection can inform incidence assays, phylogenetic studies, and prediction of future chronic liver disease burdens. In particular, it could be vital for monitoring public health progress in the context of elimination [1], as it enables the identification of ongoing transmission. From an individual-patient perspective, this information could contribute to knowledge of disease progression and assessment.

The nature of HCV transmission and its mostly asymptomatic acute infection means that the date of infection is rarely known, and there is a lack of known biomarkers available from which this information could be estimated. Accordingly, studies typically have to rely on some combination of cohort data and mathematical modelling to infer infection dates [2,3,4,5,6,7,8,9,10,11,12,13], which remain highly uncertain for most HCV-infected individuals. In the present study, we took advantage of the unique opportunity of a cohort with annual HCV screening, detailed clinical characteristics, and sampling.

A similar problem exists for human immunodeficiency virus-1 (HIV-1), which is also an RNA virus of comparable genome size that chronically infects patients. For HIV-1, it has been shown that viral diversity can be used to infer infection recency [14,15,16] and time since infection [17], and that diversity derived from next-generation sequencing (NGS) sequences is more accurate than diversity derived as the fraction of ambiguous nucleotides from Sanger sequences [18].

In this study, we aimed to investigate whether the same NGS-derived-diversity method can be applied to HCV, and to identify the region of the genome over which it is most informative to measure diversity.

## 2. Materials and Methods

### 2.1. Patients

We included 72 HCV-HIV coinfected patients from the Swiss HIV Cohort Study (SHCS), for all of whom an NGS-sequenced HCV sample was available. For patients enrolled in the SHCS, HCV-serologies are determined since 2000 every 12–24 months. Therefore, the patients considered in our study had a date of HCV infection known to within 24 months. This date of infection was calculated as the midpoint between the date of the most recent negative (RNA or antibody) test prior to the sample date, and the earliest date of either a positive HCV (RNA or antibody) test result, or the date at which the sample was taken. Patient characteristics are summarized in Table 1.

### 2.2. Sequencing

Samples were sequenced in the context of two different projects, resulting in differing, although similar, protocols. Briefly, the following sequencing protocols were used:

Project 1, as part of the Swiss-HCVree-trial (NCT02785666) [19] within the SHCS, 53 samples (see Appendix A for the description of near full-length genome sequencing): cDNA was synthesized and amplified using a one-step reverse transcription and PCR kit, or in two individual steps. Some samples then underwent a second, nested-PCR (Appendix A). Samples were pooled for sequencing, and libraries were run using MiSeq (Illumina, San Diego, CA, USA) 1 × 150 cycles.

Project 2, within the SHCS, 19 samples: Viral RNA was extracted from plasma stored in the SHCS biobank by the Nucleospin RNA Virus Kit (Macherey-Nagel, Düren, Germany). RNA was then amplified by RT-PCR in a two-step process, and some samples underwent a second, nested-PCR if necessary. HCV RNA genome sequences were generated by amplification of almost full-length HCV RNA followed by massive parallel sequencing. A MiSeq (Illumina, San Diego, CA, USA) instrument was used for sequencing with 2 × 250 bp.

Sequences were processed using MinVar version 2.2.1 (https://github.com/medvir/MinVar) [20], which filters and aligns reads before returning sequence variants. A slightly modified script was used in order to output sequence position information at the nucleotide level.

### 2.3. Diversity Score Calculation

Average pairwise diversity (APD) was calculated from nucleotide minority variant frequencies using Equation (1) [17], explanation as in [18]. This first determines whether a position has minor variants above a threshold, and subsequently sums the diversity contribution of all variants at that position. Finally, diversity across all positions is averaged. This value is functionally equivalent to the average proportion of positions at which two randomly selected sequences differ.
(1)$$\mathrm{APD}=\;\frac{1}{L}\overset{L}{\underset{i=1}{\sum }}\Theta \left(1-x_{i}^{m}-x_{c}\right)\left[\underset{\alpha }{\sum }x_{i\alpha }\left(1-x_{i\alpha }\right)\right]$$
where

*L* = length of analysed sequence.

*i* = sequence position.

*x_i_^m^* = frequency of major variant *m* at position *i*.

*x_c_* = 0.01.


$\Theta \left(x\right)=\{\begin{array}{c} 1,\;\;x>x_{c}\\ 0,\;\;x\leq x_{c} \end{array}$


*α* ∈ {A,C,G,T,deletion}.

*x_iα_* = frequency of variant *α* at position *i*.

*x_c_* is a frequency cut-off, below which variants are considered to be indistinguishable from those generated by PCR or sequencing errors. It was set to 1% as this is the approximate detection limit of a high-throughput sequencing workflow (Illumina) [21,22].

We calculated average pairwise diversity over various subsections of the HCV genome open reading frame:− All codon positions, and only the first/second/third codon positions in turn.− Whole open reading frame, individual genes, and 11 overlapping equal regions that the genome was split into (length = 502 or 501 amino acid codons).

### 2.4. Data Analysis

All results were analysed in R version 3.5.1 [23], using packages data.table [24], pROC [25], readstata13 [26], and RColorBrewer [27].

We examined how well average pairwise diversity correlated with time since infection (TSI) using linear regression. This linear model (Equation (2)) was then validated using leave-one-out cross-validation, with mean absolute error as the primary outcome. This was conducted by calculating the model coefficients using all but one sample, and then applying this model to the remaining "test" sample. The absolute value of the difference between the estimated and actual time since infection was then calculated for this test sample. This procedure was repeated for all samples in turn, and the mean value of all the absolute errors was taken, providing a simple measure by which to compare average pairwise diversity calculated across different regions.
(2)Estimated TSI = β APD + α

We studied how well average pairwise diversity could be used to infer infection recency using receiver operator characteristics (ROC) analyses, with recent infection (<1 year post-infection) defined as the positive outcome. We restricted these recency analyses to the 50 patients who could be unambiguously classified as recent (21 patients) or chronic (29 patients) due to the sample having been collected less than 12 months after the last negative HCV test or more than 12 months after the first positive HCV test.

A sensitivity analysis was conducted to assess variability between viral subtypes, as levels of diversity have been seen to vary between subtypes [28]. We therefore repeated our analyses using just those patients infected with subtype 1A, and 4D; insufficient number of samples were available from other viral subtypes for this analysis to be meaningful for them.

### 2.5. Ethics Approval and Consent to Participate

This analysis was conducted in the context of the SHCS which was approved by the ethics committees of the participating institutions (http://www.shcs.ch/userfiles/file/ethics_committee_approval_and_informed_consent.pdf, Kantonale Ethikkommission Bern, Ethikkommission des Kantons St. Gallen, Comité Départemental d’Éthique des Spécialités Médicales et de Médicine Communataire et de Premier Recours, Kantonale Ethikkommission Zürich, Repubblica et Cantone Ticino–Comitato Ethico Cantonale, Commission Cantonale d’Éthique de la Recherche sur l’Être Humain, Ethikkommission beider Basel), and written informed consent was obtained from all participants. The study protocol conforms to the ethical guidelines of the 1975 Declaration of Helsinki.

## 3. Results

### 3.1. Codon Position Makes a Notable Difference in Average Pairwise Diversity Scores

We found for the first and second codon positions a substantial correlation with time since infection but also that diversity remains low even several years post-infection (Figure 1). Conversely, average pairwise diversity over the third codon position shows a large and steady increase with time since infection, and should therefore be more informative when inferring time since infection and infection recency. However, calculating average pairwise diversity over all three positions yields similar results to considering only third-codon positions (Figure 1). This suggests that average pairwise diversity calculated over either third or all codon positions may be capable of inferring time since infection with reasonable precision.

We compared how well average pairwise diversity over individual and all codon positions could infer infection recency using ROC analyses (Figure 2). This analysis, was restricted to the 50 patients whose infections could be unambiguously classified as either recent (time since infection <1 year) or chronic (time since infection >1 year). In line with Figure 1, using third and all codon positions outperforms using first and second codon positions, supporting the notion that average pairwise diversity calculated over these positions is more informative. Due to the similarity between using the third codon and all codon positions, we continued our analyses using average pairwise diversity calculated over all codon positions, this being the easier measure to calculate as it does not require identification of the reading frame (results over the third codon position are provided in Appendix A for comparison and give very similar outcomes).

### 3.2. Restricting Average Pairwise Diversity to Certain Regions of the Genome Improves Inference of Time since Infection and Infection Recency

We calculated average pairwise diversity over the entire open reading frame of HCV, each gene individually, and for regions derived by dividing the open reading frame into eleven equal overlapping regions. We then assessed how well each average pairwise diversity scores could be used to infer infection recency (via ROC analyses) and time since infection (via linear regression and cross-validation). When comparing the three outcomes (area under the ROC curve, R^2^, and mean absolute error) for all regions across the open reading frame (Figure 3), we find that the region covering amino acid codons 503-1004– provides the best score for all three outcomes. This region spans the major 3′ part of *E2*, the entirety of *p7*, and the majority of the *NS2* genes. The regions covering the last 1000 amino acid codons of the open reading frame also perform well, but less consistently across outcomes (Figure 3). The figure also shows the HCV genes colour-coded by a composite score of all three outcomes, from which a similar pattern emerges, with *E2* and *NS2* individually performing especially well. We therefore additionally calculated average pairwise diversity over the three genes *E2*, *p7* and *NS2* combined and found similarly strong results. Appendix A provides the exact scores for all the regions tested. Appendix A shows the same analysis repeated for larger regions, which provides a similar outcome.

Based on these results, we recommend calculating average pairwise diversity using all codon positions, over the region of amino acid codons 503-1004– of the open reading frame, or using the entirety of genes *E2*, *p7* and *NS2* together. Time since infection can be estimated from these scores using Equation (2), with recommended coefficients alpha = 0.42, beta = 353.70 for average pairwise diversity over the region 503-1004–, or alpha = 0.39, beta = 320.38 for average pairwise diversity over *E2*, *p7* and *NS2*. A full list of suggested coefficients for all regions tested is available in Appendix A. Figure 4 shows the time since infection against estimated time since infection for the recommended regions.

### 3.3. Sensitivity Analysis Shows Minor Variation between Viral Subtypes

We conducted a sub-analysis including only patients infected with viral subtype 1A (50 patients, of which 33 could be included in the ROC analysis), and 4D (14 patients, of which 10 could be included in the ROC analysis), using average pairwise diversity calculated over our recommended regions. For diversity calculated over amino acid codons 503–1004, the area under the ROC curve was the same as the overall score of 0.85 for genotype 1A (0.85) and slightly lower for genotype 4D (0.67). Conversely, R^2^ was slightly higher than the overall score of 0.48 for genotype 4D (0.56), and lower for genotype 1A (0.43), whilst mean absolute error was worse than the overall 1.44 years for both genotype 1A (1.64 years) and 4D (1.52 years). A similar pattern was seen for average pairwise diversity over the genes *E2*, *p7*, and *NS2* (overall: area under the ROC curve = 0.85, R^2^ = 0.54, mean absolute error = 1.38 years. 1A: area under the ROC curve = 0.86, R^2^ = 0.51, mean absolute error = 1.51 years. 4D: area under the ROC curve = 0.62, R^2^ = 0.56, mean absolute error = 1.46 years).

## 4. Discussion

Overall, this study shows that HCV genetic diversity as calculated from nucleotide frequencies derived from NGS deep sequencing correlates with, and can be used to infer, the time since infection and infection recency. A wide range of samples of well-defined infection timepoints with differing infection durations was included, showing that this method could be employed for samples up to 16 years post-infection. With a mean absolute error of less than 1.5 years, this method can provide insight which may be particularly helpful for public health monitoring in the context of HCV elimination strategies, and the identification of recent versus chronic infections.

We additionally conducted analyses focusing on individual regions across the entire open reading frame, and have shown that the pattern of diversity differs across the genome, with some regions and codons experiencing differing levels of diversification over time. The contrast in diversity calculated over codons 1 or 2 compared to codon 3 highlights the strong effect of selection against non-synonymous mutations. It should be noted, however, that even at third-codon positions substitutions will not be completely neutral, as they can for example affect RNA secondary structure. Even though such effects may restrict the accumulation of viral diversity, they do not seem to have had a major effect on the association between time since infection and diversity as indicated by similarly strong correlations observed over the entire HCV genome (Figure 3). For example, we observed one of the strongest associations for the NS5B gene, which (in its second half) exhibits a particularly high density of regions encoding for RNA secondary structure [29,30]. Surprisingly, the most informative region was found to include the *E2* gene. As this encodes one of the two viral envelope proteins, a generally high level of diversification is unsurprising, but it was expected that selective sweeps due to immune system pressure would result in an unsteady increase in diversity unsuitable for inferring time since infection, as has been seen for the HIV envelope gene [17]. This finding defies that and suggests that other factors may be additionally influencing the accumulation of diversity within the HCV genome, causing a steadier increase.

As some variation between diversity within different viral subtypes has been reported [28], we conducted a sensitivity analysis with individual subtypes to evaluate the impact of viral subtype on our analyses. Whilst some variation was observed, overall patterns were robust across subtypes Furthermore, the largest deviation from the overall results was found for the subtype 4D ROC analyses which were conducted with only 10 samples. Our results, hence suggest that this method is applicable across different viral subtypes; however, further analysis with more samples of different subtypes is required to confirm this.

This study was limited by the number of samples available, which were all from patients already infected with HIV-1 at the time of HCV sampling, and primarily from men who have sex with men (MSM) with no recorded history of intravenous drug use (60/72 patients). The small sample size (and especially the small number of patients with large infection times) prevented the examination of non-linear effects, such as a potential saturation of diversity at large infection times, which has been observed for HIV-1 [14,18]. Furthermore, the presence of HIV-1 may affect the diversification of HCV over time, given the known impact of HIV-1 infection on HCV infection [31,32,33], and the increased virus load of HCV in the presence of HIV-1 [34,35,36]. The HCV epidemic typically consists primarily of people who inject drugs, but there is growing incidence of HCV infections among HIV-infected MSM [11,12,37,38,39,40,41]. As our study population consists almost exclusively of MSM, the validity of our results in people who inject drugs needs to be examined in future studies. Furthermore, our samples were derived from two different projects and therefore prepared and sequenced by slightly differing protocols. Whilst this variation in preparation and sequencing methods may remove some uniformity from our sample set, it more accurately reflects the scenario should the method be applied to HCV samples that have been collected and sequenced for a variety of purposes and can therefore be seen as a strength of this study.

As drug resistance testing is not routinely performed for HCV-infected patients, the availability of NGS sequences for this technique may be limited. However, as the price of NGS sequencing methods is decreasing, they are likely to be increasingly employed for surveillance purposes and gaining epidemiological understanding. Particularly in the context of elimination strategies, sequence-based information such as viral phylogenies and origins may be very desirable. Our study found a substantial correlation between time since infection and average pairwise diversity. Based on this, we could show that infection recency and time since infection can be accurately predicted by using average pairwise diversity. Thus, in the cases where samples from HCV-infected persons are sequenced, this method provides recency and date of infection information as free by-products of standard NGS sequencing, which can add to such monitoring studies.

## Figures and Tables

**Figure 1 viruses-12-01241-f001:**
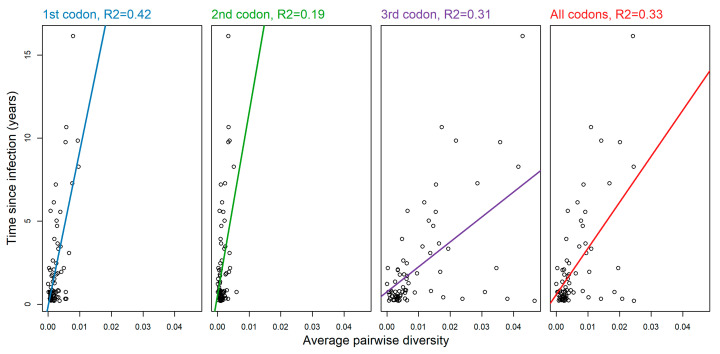
Average pairwise diversity (APD) against time since infection for APD calculated over each of the codon positions in turn, and over all three codon positions. Linear regression models are shown as solid lines.

**Figure 2 viruses-12-01241-f002:**
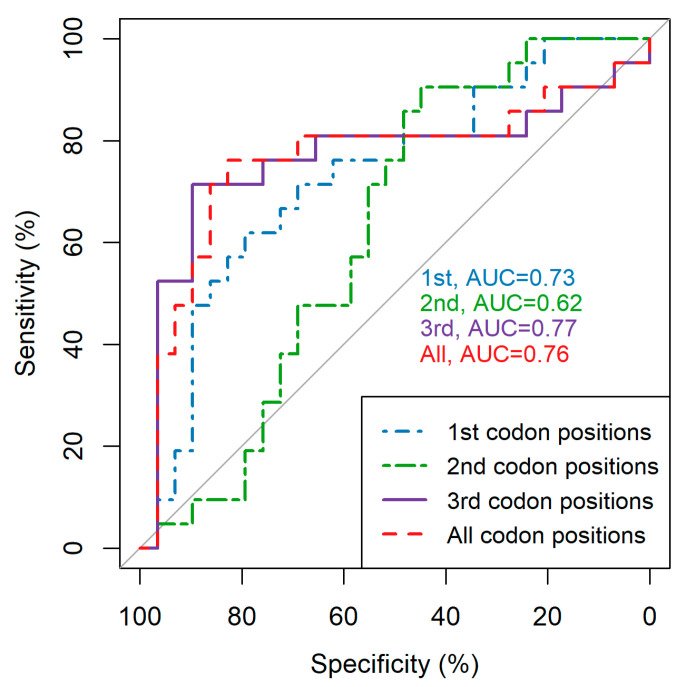
Receiver operator characteristics (ROC) curves comparing the ability of average pairwise diversity (APD) calculated over each and all codon positions to infer whether infections are recent (<1 year post-infection) or chronic. APD was calculated across the whole HCV open reading frame. All 50 patients who could be clearly classified as recent or chronic are included. Recent infection is taken as the positive outcome. AUC = area under the ROC curve.

**Figure 3 viruses-12-01241-f003:**
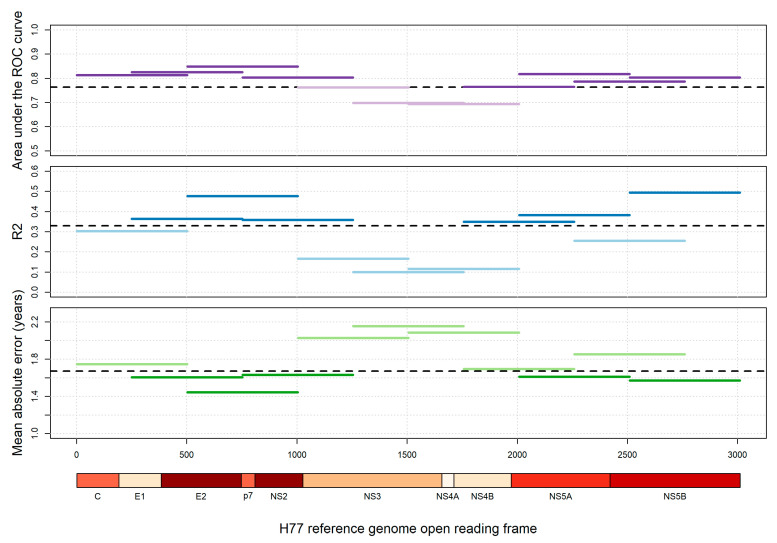
Area under the ROC curve, R^2^, and mean absolute error across the HCV open reading frame, all codon positions. The HCV open reading frame was split into 11 overlapping regions of approximately 500 amino acid codons, and average pairwise diversity (APD) was calculated over individual regions, using all codon positions. Regions were tested for their ability to categorize infection as recent (<1 year) or chronic (top), their correlation with time since infection (middle), and their ability to infer time since infection (bottom). Black dashed lines show the respective values for APD calculated over the whole open reading frame. A similar analysis was performed with diversity calculated over each gene in turn. The HCV genome is shown along the *x*-axis, with genes colour-coded for a composite (z-score sum, see Appendix A) of all three outcome scores. Darker red indicates a better overall performance. Numbers along the *x*-axis refer to amino acid positions of the H77 reference genome.

**Figure 4 viruses-12-01241-f004:**
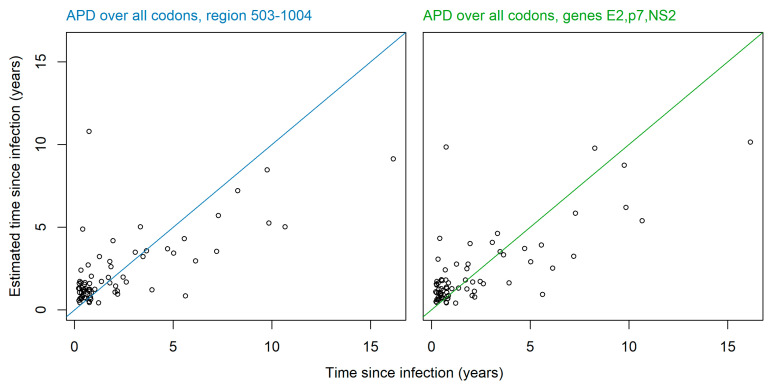
Time since infection against estimated time since infection as calculated from average pairwise diversity (APD). APD calculated over the recommended region of amino acid codons 503–1004 (**left**), and the recommended genes *E2*, *p7*, and *NS2* (**right**).

**Table 1 viruses-12-01241-t001:** Patient characteristics.

Total Number		72
Gender, *n* (%)	Female	2 (3)
	Male	70 (97)
Age when sample taken (years), median (IQR)		45 (39, 52)
Ethnicity, *n* (%)	Asian	2 (3)
	Black	2 (3)
	Hispanic	4 (4)
	White	64 (89)
HIV transmission group, *n* (%)	HET	3 (4)
	MSM	67 (93)
	IDU	1 (1)
	Unclear/unknown	1 (1)
Recorded history of intravenous drug use ever, *n* (%)	Yes	9 (13)
	No	63 (88)
HCV Viral subtype, *n* (%)	1A	50 (69)
	1B	5 (7)
	2C	1 (1)
	3A	2 (3)
	4D	14 (19)
Time since HCV infection (years), median (IQR)		0.82 (0.47, 2.5)
Clearly recent or chronic ^a^, *n* (%)	True	50 (69)
	False	22 (31)
Full coverage of gene at all codon positions, *n* (%)	*C*	69 (96)
	*E1*	69 (96)
	*E2*	70 (97)
	*p7*	70 (97)
	*NS2*	70 (97)
	*NS3*	71 (99)
	*NS4A*	71 (99)
	*NS4B*	70 (97)
	*NS5A*	69 (96)
	*NS5B*	65 (90)

IQR = Interquartile range, HET = heterosexual contacts, MSM = men who have sex with men, IDU = injection drug use. ^a^ Sample collection less than 12 months after last negative HCV test or more than 12 months after first positive HCV test.

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
