# Peer review of "HCV Genetic Diversity Can Be Used to Infer Infection Recency and Time since Infection"

_viruses, 2020, doi:10.3390/v12111241_

Round 1

Reviewer 1 Report

The study presents genetic distance calculations of HCV isolates. The methods appear appropriate (given that my expertise is more virological than bioinformatic), and the results support the conclusions. I recommend publication of this manuscript after revision. 

I have some concerns: 

  1. In Figures 1 and 4, the regression lines appear - intuitively - not to perfectly represent the distribution of data points. Could a non-linear regression lead to better fit and higher R2 values? That should be checked and discussed. 
  2. In the entire genome, in particular in the second half of the NS5B coding region, RNA secondary structures may pose bias on the codon selection (Fricke et al., "Global importance ...", Bioinformatics, 2019). Does this have an influence on the output in Fig. 3? This should be checked and discussed. 

Minor points: 

  • some citations are shown with an error code. 
  • - citation No. 18 is named "Manuscript in preparation". That should be removed and explained in the main text as unpublished results. 
  • In the abstract, two blanks preceeding brackets are missing

Reviewer 2 Report

Major comments

This is an interesting paper focus on usefulness of HCV diversity as tool for prediction of recency of the infection, data especially helpfulness for epidemiological aims as screening of contacts of risk.

The main concern about the article is that, according to the methods, the number of patients whose HCV infection can be catalogued as acute or infection is 50 patients. However, in the abstract it is mentioned that the number of included patients were 72. The real number of included subjects should be clarified.

Throughout the manuscript there are many typos regarding references (page 3, line 83, page 5, line 157, 162, 154, 185…). Moreover, written English may be revised, especially in the results (pages 5-7).

Minor comments

Results

Page 5, paragraph “Codon position makes a notable difference in average pairwise diversity scores”: the first 2 sentences may be obviated since they have been mentioned in the “introduction”.

Discussion

It may be useful the addition of a conclusion paragraph at the end of the discussion summarizing the most relevant findings of the manuscript.

Round 2

Reviewer 1 Report

The authors have addressed my concerns appropriately. 

Reviewer 2 Report

The authors have successfully addressed all the issues suggested by the reviewers. Congrats for your work.